# DEPEND study protocol: early detection of patients with pancreatic cancer – a pilot study to evaluate the utility of faecal elastase-1 and $^{13}$C-mixed triglyceride breath test as screening tools in high-risk individuals

Paul Remi Afolabi [1,2] Declan McDonnell,[1,3] Christopher D Byrne,[1,3] Sam Wilding,[4] Victoria Goss,[4] Jocelyn Walters,[4] Zaed Z Hamady[3,5]

For numbered affiliations see end of article.

**Correspondence to**
Zaed Z Hamady;
Z.Hamady@soton.ac.uk

## ABSTRACT

**Introduction** Pancreatic cancer (PC) is the fifth leading cause of cancer-related death in the UK. The incidence of PC is increasing, with little or no improvement in overall survival and the best chance for long-term survival in patients with PC relies on early detection and surgical resection. In this study, we propose the use of a diagnostic algorithm that combines tests of pancreatic exocrine function (faecal elastase-1 (FE-1) test and the $^{13}$C-mixed triglyceride ($^{13}$C-MTG) breath test) to identify patients with PC that urgently needs imaging studies.

**Methods and analysis** This prospective pilot (proof of concept) study will be carried out on 25 patients with resectable PC, 10 patients with chronic pancreatitis and 25 healthy volunteers. This study will construct a predictive algorithm for PC, using two tests of pancreatic exocrine function, FE-1 test and the $^{13}$C-MTG breath test. Continuous flow isotope ratio mass spectrometry in the $^{13}$C-MTG breath test will be used to analyse enriched $^{13}$CO$_2$ in exhaled breath samples. The additional predictive benefit of other potential biomarkers of PC will also be considered. Potential biomarkers of PC showing abilities to discriminate between patients with PC from healthy subjects or patients with chronic pancreatitis will be selected by metabolomic analysis.

**Ethics and dissemination** The study was approved by the North of Scotland Research and Ethics Committee on 1 October 2020 (reference: 20/NS/0105, favourable opinion granted). The results will be disseminated in presentations at academic national/international conferences and publication in peer-review journals.

---

**Strengths and limitations of this study**

► We will use a non-invasive $^{13}$C-breath test to construct a predictive algorithm for detecting pancreatic cancer (PC).

► PC patients with potentially curative pancreatic resection will be recruited in this study.

► The predictive algorithm may have poor ability to discriminate between patients with PC and chronic pancreatitis.

► Participants' return of faecal specimens may be limited by embarrassment, hesitancy or inability of the study participants to provide a sample.

---

Since complete surgical resection remains the only potentially curative option for PC, it is crucial to identify PC at an early stage, when it is still resectable. Unfortunately, only 10%–20% of patients with PC are diagnosed at a stage when curative surgery remains an option.[2]

The early symptoms of PC are usually vague and include weight loss, upper abdominal pain, backache, fatigue, diarrhoea and nausea. In 93% of patients, these symptoms may have been present for up to 2 years before the diagnosis of PC.[3] Recent studies show that around 40% of patients diagnosed with PC require three or more visits to their general practitioner (GP) before they are referred to a specialist[4 5] thereby leading to a delay in the diagnosis. There is also an increased risk of developing PC in those with new-onset diabetes mellitus (NODM) that is, less than 3 years since diagnosis[6] and as many as 25% of PC cases are believed to present with NODM.[7]

Recent survey of GPs commissioned by Pancreatic Cancer Action[8] suggested that the

## INTRODUCTION

Pancreatic cancer (PC) is the fifth leading cause of cancer-related death in the UK. The UK has one of the worst survival rates in Europe, with an average life expectancy at diagnosis of just 4–6 months and 20% survival at 1 year. In addition, in the UK, only about 7% of people survive for 5 years or longer.[1]

delay between not having access to a reliable diagnostic referral pathway and a confirmed diagnosis of PC is a major barrier to the early detection of PC. As the symptoms of PC are often non-specific and unlikely to prompt further clinical investigation, there is an urgent need to have a reliable and cheap screening test that can be used to identify high-risk individuals who require urgent imaging studies such as CT scan to confirm or exclude the presence of a pancreatic mass.[9] Having such a test with reliable sensitivity and negative predictive value (NPV), would enable the GP to select patients for urgent referral to further investigations of PC. A high NPV value would enable the GP to reassure the patient that they did not need further imaging studies. Our innovative approach is to use tests of pancreatic exocrine function combined as a screening test in patients presenting with the vague symptoms that may be associated with PC.

Although pancreatic exocrine insufficiency (PEI) is a well-known complication of PC,[10 11] its presence is frequently overlooked in patients with advanced PC resulting in a decreased quality of life, malnutrition and morbidity.[12] Currently, the diagnostic tests of pancreatic exocrine function such as faecal elastase-1 (FE-1) or the $^{13}$C-mixed triglyceride ($^{13}$C-MTG) breath test, may be used to detect PEI in patients after pancreatic surgery and chronic pancreatitis.[13–16]

FE-1 test is a simple indirect and non-invasive method for assessing pancreatic secretion, which measures faecal concentrations of elastase-1, a proteolytic enzyme produced exclusively by pancreatic acinar cells, which binds to bile salts and passes through the gut with minimal degradation. FE-1 level correlate well with output of other pancreatic enzymes, such as amylase, lipase and trypsin[17 18] and have been shown to detect PEI in patients with advanced PC and following pancreatic surgery.[14–16]

The $^{13}$C-MTG breath test assesses pancreatic lipase, it involves the ingestion of a $^{13}$C-labelled mixed triglyceride (1,3-distearoyl, 2-(carboxyl-$^{13}$C)-octanoyl glycerol) mixed with a test meal. The $^{13}$C-MTG contains a $^{13}$C-labelled medium-chain fatty acid ($^{13}$C-octanoate) at the Sn-2 position, and two long-chain fatty acids (stearic acid) at the Sn-1 and Sn-3 positions of the glycerol backbone of the triacylglycerol. The ingested $^{13}$C-MTG then undergoes the lipolysis of the two-long chain fatty acids from $^{13}$C-MTG by pancreatic lipase to produce free fatty acids (stearic acid and $^{13}$C-octanoate) and monoacylglycerol. After the intestinal absorption, the $^{13}$C-free fatty acids are oxidised in the liver producing $^{13}CO_2$, which is then released on breath. Therefore, the $^{13}CO_2$ concentration in breath reflects pancreatic lipase activity. The $^{13}$C-MTG breath test can be an alternative or complementary to the FE-1 test for PEI. The $^{13}$C-MTG breath test has been validated in patients with chronic pancreatitis[19] and patients with PC following surgical resection of the cancer,[20 21] but not prior to surgery in patients with resectable PC.

Therefore, in order to improve the prognosis of patients with PC, it is essential to detect tumours at early stages, when they are more likely to be resectable. In this

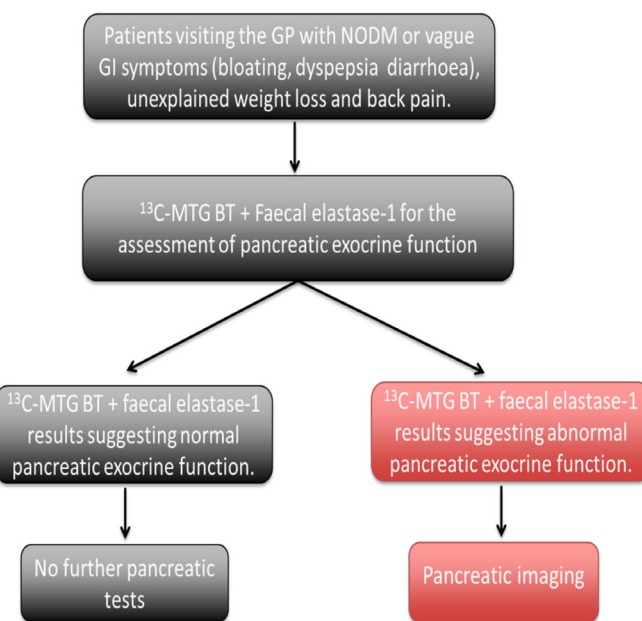

**Figure 1** Proposed diagnostic tool consisting of faecal elastase-1 test and the $^{13}$C-mixed triglyceride breath test ($^{13}$C-MTG BT) to screen for pancreatic exocrine dysfunction in patients 'at-risk' of pancreatic cancer. GI, gastrointestinal; GP, general practitioner; NODM, new- onset diabetes mellitus.

study, we will validate FE-1 test and $^{13}$C-MTG breath test as potential early diagnostic biomarkers of resectable PC and test their performance in discriminating between patients with PC and healthy subjects. These tests can provide a non-invasive risk stratification tool to identify patients at risk of having PC when they present with NODM, weight loss or vague upper gastrointestinal symptoms. Furthermore, this will provide a rational approach to patients identified to have PEI to be selected to have pancreatic imaging such as CT scan to confirm the presence of PC and having a pancreatic resection earlier (figure 1). In addition, we will also be identifying other potential biomarkers of PC through unbiased multiomics analysis of blood samples (proteomics and metabolomics). The resulting data will be compared with the healthy control group and patients with chronic pancreatitis aiming to identify signals that are distinct to PC.

## METHODS AND ANALYSIS
### Main centre
Southampton NIHR-Biomedical Research Centre, University Hospital Southampton NHS Foundation Trust, Southampton, UK.

### Dates of the study
From 16 March 2021 to 16 March 2022.

### Design
This is a single cross-sectional pilot study of patients with PC, chronic pancreatitis and healthy controls, matched for age and sex. The primary objective is to determine if

$^{13}$C-MTG breath test and FE-1 concentrations combined can discriminate between the groups. Patients with PC will be recruited at the preoperative stage within 2 weeks of initial diagnosis. The secondary objective of this study is the analysis of multiomic (proteomic and metabolomic) profiles and comparison between the groups.

### Inclusion criteria

At the University Hospital Southampton NHS Foundation Trust, on average, 120 patients with PC are treated with surgery while 350 patients with PC are treated with chemotherapy every year. Male and female participants aged from 30 to 85 within the Wessex PC catchment area will be suitable for inclusion. In the cancer cohort, patients with a diagnosis of resectable pancreatic ductal adenocarcinoma (PDAC) (stage 1–2) will be included. Patients with chronic pancreatitis and healthy subjects will also be recruited as control groups and matched for age and sex to the cases of PDAC.

### Exclusion criteria

Patients and controls with autoimmune or other chronic inflammatory conditions, or those who have previously had pancreatic disease or liver dysfunction will be excluded.

### Intervention(s) or method

Each participant will be given a participant information sheet explaining the research and will sign an informed consent form. Participants are required to fast overnight for at least 12 hours prior to undergoing the $^{13}$C-MTG. At the Clinical Research Facility, full infection control precautions (related to COVID-19 pandemic) will be adhered to throughout the visit. Participants will have their temperature checked on arrival to ensure they are not feverish as part of screening for COVID-19, which will exclude them from the study. Anthropometric measurements (height, weight and waist circumference) of the participants will be recorded. Baseline blood samples will be taken prior to undergoing the $^{13}$C-MTG breath test. The data collected will include the following: full blood count, urea and electrolyte blood test, liver function tests (aspartate aminotransferase (AST), alanine aminotransferase (ALT), Alkaline phosphatase (ALP), albumin and Gamma-glutamyl transferase (GGT))/clotting studies, glycated haemoglobin (HbA1c %), lipid profile, vitamins and micronutrients, hyaluronic acid, procollagen III amino terminal peptide and tissue inhibitor of metalloproteinase-1. These tests will be analysed immediately after the blood has been drawn from the participants by the chemical pathology service laboratory at Southampton General Hospital. Further blood samples will be processed and stored for multiomics analysis. Participants will also be asked to provide spot stool samples for FE-1 concentrations by using a commercially available ELISA kit. Results of the FE-1 test will be expressed as μg/g of stool.

### $^{13}$C-MTG breath test protocol

Each participant will be asked to provide two baseline breath samples by blowing into 2×10 mL exetainer tubes through a straw while wearing a facemask. After that, participants will be given an oral dose of 250 mg of 2-[$^{13}$C]-octanoyl-1,3-distearin ($^{13}$C-MTG-Cambridge Isotope Laboratories, Andover, Massachusetts, USA) together with a solid test meal consisting of a crispbread or other gluten-free alternative, and 200 mL of water. Postprandial breath samples will be then collected every 30 min for 4 hours. No food or drinks except for water will be allowed during the duration of the breath test study. The collected breath samples will be analysed for the ratio of $^{13}CO_2$ to $^{12}CO_2$ using a Continuous Flow Isotope Ratio Mass Spectrometer (CF-IRMS SERCON Ltd, Crewe, UK) at the Southampton Centre for Biomedical Research. The increase in $^{13}CO_2$ content with regard to the baseline value of the initial breath sample is expressed as atom percentage excess and we will express the results of the $^{13}$C-MTG breath test as the cumulative percentage of $^{13}$C-label recovered on breath as $^{13}CO_2$ over 4 hours (cumulative percentage dose recovered (cPDR) over 4 hours).

### Primary outcome

Prediction of PC using a logistic model with measurement of FE-1 concentration expressed as μg/g of stool and $^{13}$C-MTG breath test expressed as cPDR over 4 hours as covariates.

### Statistical analysis and data analysis plans
#### Sample size calculation

The sample size calculation for the proposed study is based on a previous study assessing pancreatic exocrine function using a $^{13}$C-triglyceride breath test in healthy subjects and patients with a localised pancreatic mass.[13] In this study, they used the $^{13}$C-trioctanoin breath test to assess pancreatic exocrine function in 14 patients with a localised pancreatic mass and five healthy control subjects. The results of the study showed means±SD of the recovery of $^{13}CO_2$ over 3 hours after undergoing the $^{13}$C-trioctanoin breath test of 42%±3.4% for the healthy controls and 24.2%±10.5% for patients with a localised pancreatic mass with an effect size or mean difference between both groups of 17.8% and a within group SD of 9.1% (based on the SD estimates of 3.4% and 10.5% for each group, respectively).

Using data from this study, we calculated that a sample size of 25 subjects in each group (patients with PC, patients with chronic pancreatitis and healthy control subjects) will give 100% power to detect a difference of 17.8% between patients with PC and healthy control group assuming an SD of 10.5% in patients with PC at a significance level of 5% using a two-tailed test. The power calculation was carried out using IBM SPSS SamplePower V.3.

## Statistical tests

Data will be presented as mean+SD, median+IQR when appropriate and 95% CI. Comparison of results of FE-1 and the $^{13}$C-MTG breath test between each group will be evaluated by employing either the one-way analysis of variance test (if the variables are normally distributed and parametric validity conditions are fulfilled) or the Mann-Whitney U test or Kruskal-Wallis test (if the variables are not normally distributed and non-parametric validity conditions are fulfilled). A multi-variable logistic regression analysis will be performed in order to determine the independent factors that are significantly associated with the presence of PDAC and logistic regression models will be constructed based on the identified independent factors. Furthermore, multi-variable regression analysis will be used to determine the optimal algorithm that combines the results of FE-1 test and $^{13}$C-MTG breath test. We will also test whether potential biomarkers of PC identified by unbiased omics analysis improve the prediction of PDAC. A receiver operating characteristic curve will be constructed to evaluate the diagnostic accuracy (sensitivity, specificity, positive and NPV, receiver operator curve analysis) of the algorithm for PDAC. Furthermore, the optimal cut-off point for the algorithm to identify patients with PC will be estimated using the Youden index. A two-sided p value of $<0.05$ will be considered to indicate statistical significance. All analysis will be performed using SPSS (IBM, V.25.0) or Stata (Statacorp, V.16.0).

## Patient and public involvement statement

Patients, healthy subjects and the public were not involved in the study design, recruitment and conduct. We do not plan to inform the result to the study participants unless they apply for it.

## ETHICS AND DISSEMINATION

Ethical approval has been granted by the North of Scotland Research and Ethics Committee, reference: 20/NS/0105, 1 October 2020. This manuscript reflects the latest protocol V.7 approved 29 October 2020.

The study will be conducted according to the principles of Good Clinical Practice, General data Protection Regulation and Data Protection Act 2018 for Health and care Research. The sponsor and study team will ensure approval of the study protocol, participant information sheets, consent forms, letters to general practitioners and supporting documents by the appropriate regulatory body and research and ethics committee prior to participant recruitment. Documents will be stored securely with restricted access for at least 10 years.

Written, informed consent will be obtained from each patient and an identification number provided. Any published data will not contain personal identifiable data. The results of this study will be disseminated by presentation at academic national/international conferences and publication in peer-review journals.

**Author affiliations**
$^1$Human Development and Health, University of Southampton Faculty of Medicine, Southampton, UK
$^2$Southampton Centre for Biomedical Research Mass Spectrometry, NIHR Southampton Biomedical Research Centre, Southampton, UK
$^3$Southampton Centre for Biomedical Research, NIHR Southampton Biomedical Research Centre, Southampton, UK
$^4$Southampton Clinical Trials Unit, University Hospital Southampton NHS Foundation Trust, Southampton, UK
$^5$HPB Unit, University Hospital Southampton NHS Foundation Trust, Southampton, UK

**Acknowledgements** The authors would like to thank all the patients and healthy subjects who participated in this study.

**Contributors** ZH and PRA have designed the study concept and formulated the initial idea. ZH is the chief investigator and responsible for protocol review. DM and PRA wrote the initial protocol. CDB, PRA, SW and ZH were involved in statistical analysis plan. DM and ZH were responsible for organising the study operation requirements, identifying and discussion with eligible participants. DM is responsible for patient recruitment and consent. DM and PRA prepared the test meal and obtained the samples from the participants, VG is sorting the standard operating procedures (SOPs) and logistics with biobanking and sample process. PRA is responsible for the analysis of the breath samples and the interpretation of the results from the breath test. JW is managing the study team and logistics related to clinical trial involvement. PRA, DM, CDB, SW, VG, JW and ZH were involved in drafting of the protocol.

**Funding** This work is supported by Cancer Research UK (grant number C45617/A29908). DM partly funded by NIHR-BRC fellowship grant number NIHR-INF-0953.

**Competing interests** None declared.

**Patient and public involvement** Patients and/or the public were not involved in the design, or conduct, or reporting, or dissemination plans of this research.

**Patient consent for publication** Not applicable.

**Provenance and peer review** Not commissioned; externally peer reviewed.

**ORCID iD**
Paul Remi Afolabi http://orcid.org/0000-0002-0553-1578

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
