## [Reviewer comments · BMJ Open]

ARTICLE DETAILS

TITLE (PROVISIONAL)	DEPEND study protocol: Early detection of patients with pancreatic cancer: a pilot study to evaluate the utility of faecal elastase-1 and 13C-mixed triglyceride breath test as screening tools in high-risk individuals.
AUTHORS	Afolabi, Paul; McDonnell, Declan; Byrne, C. D.; Wilding, Sam; Goss, Victoria; Walters, Jocelyn; Hamady, Zaed

VERSION 1 – REVIEW

REVIEWER	Eithne Costello-Goldring University of Liverpool, molecular and clinical cancer medicine
REVIEW RETURNED	23-Oct-2021

GENERAL COMMENTS	This is an interesting study in a very important research area. The title of the study indicates early detection of patients with pancreatic cancer. The study will recruit patients with resectable pancreatic cancer. The authors should define what they mean by early. Do they mean patients who are diagnosed with resectable pancreatic cancer? It would be good to make this clear earlier in the manuscript, for example in the Introduction. The introduction states that only 3% of people survive for 5 years or longer and this figure has not improved four decades. Please update these numbers in line with the current overall five-year survival of ~9%. On page 12, line 8, the authors state that “the sample size collection for the proposed study is based on a previous study at assessing pancreatic exocrine function using a 13C-triglyceride breath test in healthy subjects and patients with a localised pancreatic mass. Although the reference is provided, it would be helpful if the authors provided the data from this study that was used to inform the power calculation.
--

REVIEWER	Matthias Löhr Karolinska Institute, Matthias Löhr
REVIEW RETURNED	11-Nov-2021

GENERAL COMMENTS	The study protocol outline by Afolabi and others from Southampton relates to an interesting approach how to diagnose patients with suspected PC
---

VERSION 1 – AUTHOR RESPONSE

Reviewer: 1

Dr. Eithne Costello-Goldring, University of Liverpool Comments to the Author:

This is an interesting study in a very important research area.

The title of the study indicates early detection of patients with pancreatic cancer.

The study will recruit patients with resectable pancreatic cancer.

-The authors should define what they mean by early. Do they mean patients who are diagnosed with resectable pancreatic cancer? It would be good to make this clear earlier in the manuscript, for example in the Introduction.

Reply: We have defined early detection in patients who are diagnosed with resectable cancer. We mentioned the definition of early in the first paragraph of the introduction and we have now added a sentence (page 6, lines 124-125) to confirm the early detection of tumors in patients with resectable pancreatic cancer.

-The introduction states that only 3% of people survive for 5 years or longer and this figure has not improved four decades. Please update these numbers in line with the current overall five-year survival of ~9%.

Reply: We have now updated the current overall five-year survival rate of ~7% in the UK with a new reference in the Introduction section on page 4, lines 62-63.

-On page 12, line 8, the authors state that "the sample size collection for the proposed study is based on a previous study at assessing pancreatic exocrine function using a ¹³C-triglyceride breath test in healthy subjects and patients with a localised pancreatic mass. Although the reference is provided, it would be helpful if the authors provided the data from this study that was used to inform the power calculation.

Reply: We have now added an extra paragraph in the Statistical analysis and data analysis plans section on page 11, lines 215-222 which provides the original result from the study that we used to inform our sample size power calculation.

Reviewer: 2

Dr. Matthias Löhner, Karolinska Institute Comments to the Author:

The study protocol outline by Afolabi and others from Southampton relates to an interesting approach how to diagnose patients with suspected PC